# SPIDER: Boosting Blind Face Restoration via Simultaneous Prior Injection and Degradation Removal

## Abstract

Existing blind face restoration (BFR) methods suffer from drastic performance drop under severe degradations. A common strategy is to first remove degradations and then restore the face by fully harnessing generative prior. However, this sequential pipeline risks discarding subtle but crucial cues from already limited low-quality (LQ) inputs. To address this, we ingeniously introduce a new learning paradigm: simultaneous prior injection and degradation removal (SPIDER). Unlike existing approaches, SPIDER injects semantic prior before degradation removal, thereby preserving identity-relevant features and mitigating the impact of corrupted LQ features. SPIDER consists of two key modules: (1) a prior injection module that distills purified degradation-unaware semantic control tokens from vision-language models, and (2) a degradation removal module equipped with an image-to-text degradation mapper and a degradation remover that refines distorted features into robust representations. Extensive experiments on both synthetic and real-world datasets, including challenging surveillance scenarios, demonstrate SPIDER's clear superiority over state-of-the-art BFR methods.

## 1 Introduction

Blind Face Restoration (BFR) is a challenging task that aims to recover high-quality (HQ) face images from low-quality (LQ) ones that suffer from unknown and complex degradations such as low resolution [3, 6], blur [48], noise [13, 28], and JPEG compression [5]. This is an inherently ill-posed problem as the information loss caused by the degradations leads to an overwhelming number of plausible HQ solutions consistent with the same LQ input. To mitigate the ill-posedness, recent studies have explored various prior-based methods to produce high-fidelity outputs.

As illustrated in Figure 1, existing prior-based BFR methods fall into three main paradigms: 1) Continuous generative prior (e.g., GFPGAN [36], which learns accurate latent codes via GAN inversion to reconstruct HQ faces with high fidelity; 2) Discrete generative prior (e.g., Codeformer [52], DAEFR [30]), which uses vector quantization to map degraded inputs into semantic tokens and harness a fixed HQ codebook for high-quality restoration; and 3) Diffusion-based conditional generation (e.g., DiffBIR [18], FaithDiff [2]), which reframes the restoration into conditional generation employing the powerful expressiveness of diffusion prior to achieve significant improvements in fine detail, perceptual fidelity, and overall realism. Many SOTA methods [18, 2, 37, 41] belong to the third paradigm and achieve promising results on mild to moderate degradations. However, under severe or extreme degradation, whether synthetic or real-world, they often introduce artifacts or even fail catastrophically in the results, as demonstrated in Figure 2. Taking extreme surveillance degradations in the fourth row as an example, since aliasing and jagged artifacts are not presented in the synthesized training data, the existing models mistakenly use the corrupted signals as the actual

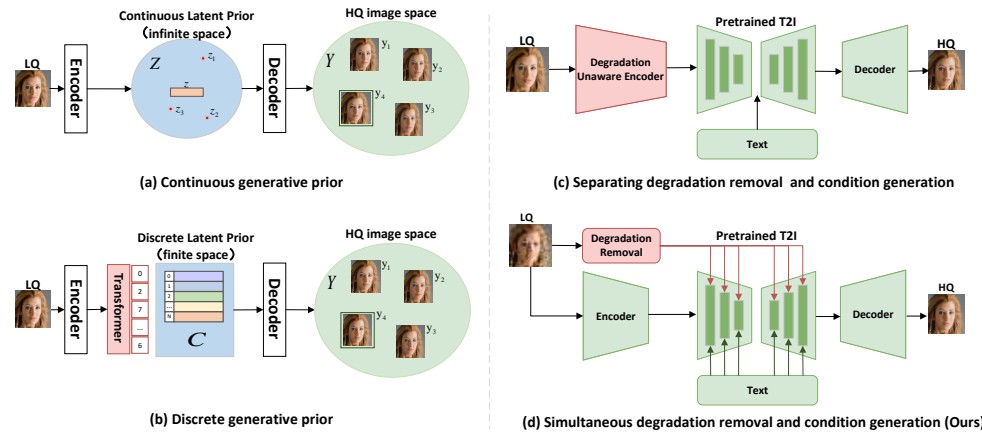

Figure 1: Comparison with existing paradigms for blind face restoration.

features, resulting in erroneous details in the results. We argue that the noises induced by the severe degradations are the primary cause of restoration failure. To address this, many methods [18, 37] first remove degradations explicitly or implicitly and then restore the face by leveraging powerful generative priors (Figure 1 (c)). However, this sequential pipeline risks discarding subtle but crucial cues from already limited LQ inputs, further elevating the ill-posedness of BFR.

To overcome this, we ingeniously propose Simultaneous Prior Injection and Degradation Removal (SPIDER), a new learning paradigm to enhance face restoration (Figure 1 (d)). Rather than removing degradation first, SPIDER injects semantic prior (i.e., combined with diffusion prior) before degradation removal. Intuitively, this design not only enriches the representation of relevant facial content but also amplifies both signal and noise. This amplification enables the subsequent degradation removal module (DRM) to more effectively differentiate between informative structures and unwanted noise, resulting in substantially improved restoration fidelity (Figure 2).

Specifically, SPIDER consists of two key components. The Prior Injection Module distills degradation-unaware semantic tokens using a vision-language model (VLM), such as LLaVA [19], to generate rich textual descriptions from degraded images. These semantic priors are subsequently injected into the diffusion generation pipeline at multiple levels, providing robust and context-aware guidance. The DRM comprises an image-to-text degradation mapper and a degradation remover, which together project noisy visual representations into a purified textual embedding space aligned with the injected prior. This design leverages the noise-resilience of the textual modality and performs degradation filtering through semantic alignment, which is more robust to perturbations than direct visual-space restoration. By jointly integrating semantic prior injection and degradation removal via our proposed decoupled cross-attention (DCA) mechanism, SPIDER delivers state-of-the-art restoration results under severe degradations in both synthetic and real-world scenarios.

SPIDER achieves state-of-the-art results on both existing synthetic and real-world benchmarks and our newly introduced SCface dataset [8] of extreme surveillance face images. Beyond its superior BFR performance, SPIDER pioneers a novel learning paradigm *injecting prior before degradation removal* that can be extended to a wide range of restoration tasks beyond blind face restoration.

## 2 Related Work

### 2.1 Blind Face Restoration

Recent BFR approaches mainly leverage generative prior to reconstruct faces with high realism and faithful details. Representative latent-prior-based methods such as GFPGAN [36] and GPEN [44] encode LQ face images into semantically faithful latent codes, enabling faithful reconstruction of their HQ counterparts using StyleGAN-based generative prior [11]. Despite improvements in fidelity, these methods often introduce artifacts when the input images exhibit complex degradations not covered by the training data. State-of-the-art methods like Codeformer [52], RestoreFormer [39],

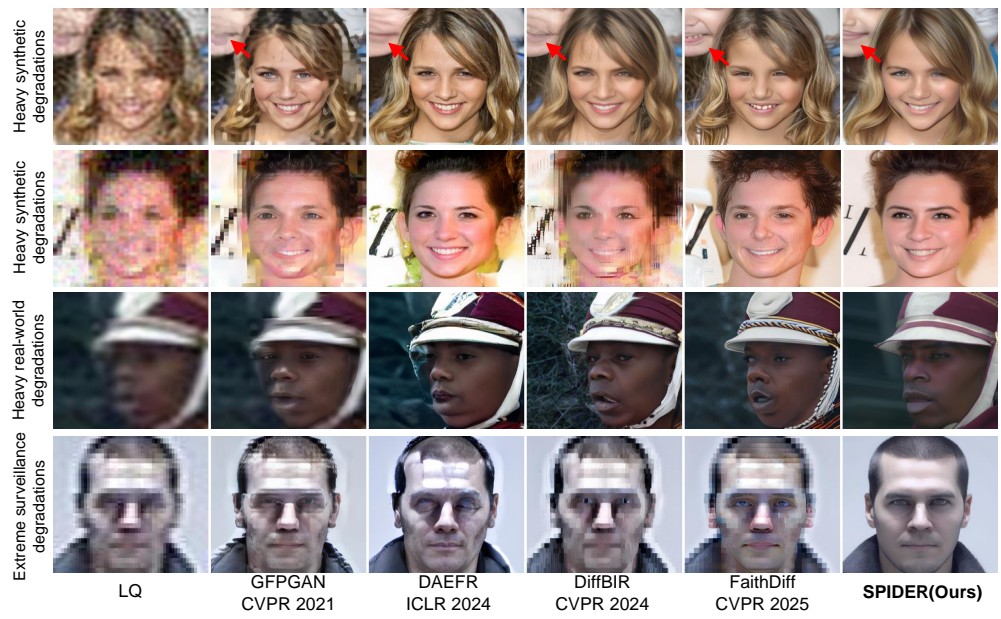

Figure 2: Comparisons with representative face restoration approaches on both synthetic (CelebA-Test [10]) and real-world (WIDER [42], SCface [8]) LQ images under various degradations
.

and DAEFR [30] utilize discrete HQ codebook to generate high-fidelity face details by exploiting Vector-Quantized (VQ) dictionary learning [7, 31]. However, the fixed-size codebook inherently limits the expressiveness ability of such discrete prior, which can hinder the faithful reconstruction of diverse and complex facial structures.

Recent works [45, 46, 34, 18, 2] reframe face restoration as conditional image generation using powerful diffusion prior, significantly advancing BFR quality. VSP [22] introduces prompt-based inference to further refine restoration results. StableSR [34] and TASR [15] finetune the temporal embedding layers to produce time-aware features that adaptively modulate features across the denoising steps. DiffBIR [18] and DR2 [37] perform degradation removal and conditional image generation sequentially: they first remove degradation using an off-the-shelf model, and then refine details. PASD [45] and SUPIR [46] enhance LQ feature extraction with stronger encoders. The latest work FaithDiff [2], employs BSRNet [47] for initial restoration and extracts text embeddings via LLaVA [19]. It further improves this paradigm by jointly training the encoder and diffusion model in an end-to-end fashion, enabling their synergistic evolution and enhancing alignment between the extracted features and the generated content.

Although the above methods have demonstrated strong performance in restoring faces under moderate degradations, they often struggle in real-world scenarios involving severe and complex degradations. This results in visual artifacts, structural distortions, and semantic inconsistencies. A key challenge lies in the model's difficulty in distinguishing intrinsic, reliable facial features from degradation-induced noise. Consequently, synthesizing faces from corrupted or noisy representations can lead to erroneous or unrealistic restoration outcomes. Therefore, effectively removing degradations is a prerequisite for achieving faithful and high-quality restoration.

## 2.2 Degradation Removal in Blind Image Restoration

Recent blind image restoration methods increasingly focus on learning degradation processes to enhance realism and adaptability. Due to the limitations of handcrafted degradation assumptions, AND [35] introduces an adversarial degradation generator that synthesizes pseudo-degraded images, thereby bridging the domain gap between synthetic and real-world degradations in supervised restoration. DiffBIR [18] and FaithDiff [2] both adopt a two-stage design, where degradation

is first removed and then image quality is refined. TextualDegRemoval [17] leverages textural modality representations to generate clean guidance images under natural degradations such as rain and snow, using them as reference images to enhance blind image restoration. However, despite differences in representation and guidance modality, these methods share a structurally decoupled architecture: degradation removal is treated as a prerequisite, independent of the image generation process. This decoupling hinders joint optimization and frequently leads to unstable outputs when handling occlusion, motion blur, or structural corruption—especially in face restoration, where identity consistency and semantic fidelity are particularly fragile.

## 2.3 Vision-Language Models

Vision-language models (VLMs) have advanced rapidly with CLIP [26] providing strong semantic prior by aligning image and text embeddings. DA-CLIP [23] models image content and degradations jointly, enabling multi-task restoration. SSP-IR [51] enhances geometric consistency by integrating structural contour information into CLIP-based prior, while FUSION [21] improves cross-modal understanding through deep feature fusion. Apart from general VLMs, researchers have also proposed vision-language architectures specifically designed for face images. FCLIP [4] uses dual-branch learning on FaceCaption-15M for better attribute alignment. Face-MLLM [29] employs a three-stage strategy on a large-scale QA dataset to enhance fine-grained attribute reasoning and instruction following. FaceInsight [14] integrates keypoint detection and attention mechanisms to ensure structural and identity consistency. However, these methods remain unpublished or proprietary, hindering the integration of face-oriented VLMs into BFR task.

## 3 Proposed Method: SPIDER

### 3.1 Framework overview

As illustrated in Figure 3, our SPIDER restores HQ face images from their LQ counterparts by simultaneously injecting cross-modal semantic prior and removing degradations. The training process of SPIDER is divided into two stages. In Stage I (Figure 3(b)), we train a degradation removal module (DRM) to remove degradations at the textual level, where the image content and degradation information are loosely coupled, making it more effective to isolate and remove noise. In Stage II (Figure 3(a)), we employ a large vision-language model (i.e., LLaVA) to generate detailed text descriptions of HQ face images, enabling the extraction of fine-grained semantic prior. Meanwhile, the pretrained DRM is used to "erase" feature corruption at multiple scales. These two branches (i.e., prior injection and degradation removal) interact via a decoupled cross attention (DCA) mechanism (Figure 3(c)) integrated into each block of both the UNet [27] and ControlNet [49]. This collaborative design ensures robust guidance and effective noise suppression, ultimately leading to faithful HQ face restoration.

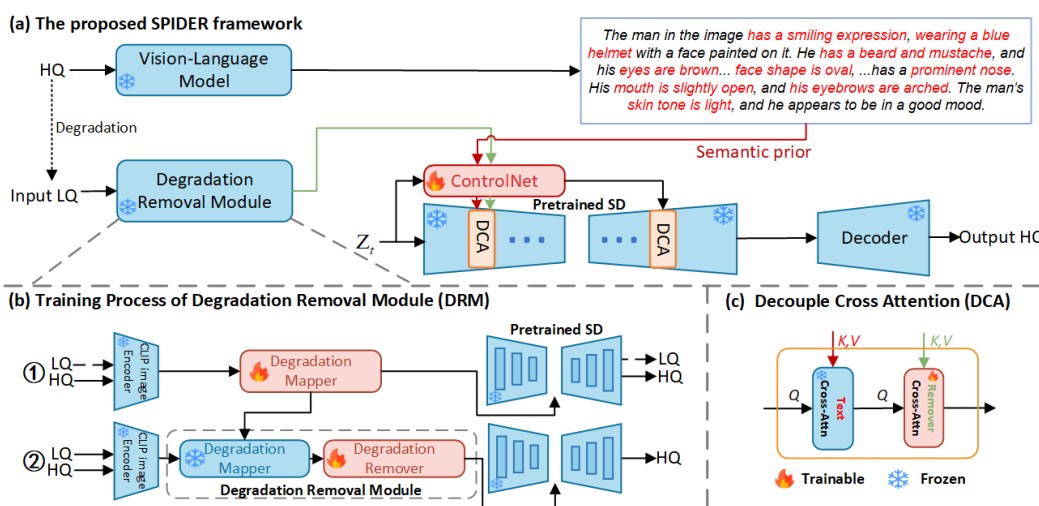

Figure 3: Framework of our proposed SPIDER model.

## 3.2 Semantic Prior Injection

Many recent super-resolution models leverage prior knowledge provided by vision-language models, such as FaithDiff [2], XPSR [25], and AuthFace [16]. The performance of LLaVA is highly dependent on well-crafted prompts [20]. Considering the highly structured nature of human faces, we modify the prompt design to focus the output on detailed facial component structures. Experiments in XPSR [25] and SPIRE [24] demonstrate that LLaVA can implicitly infer both the type and severity of noise artifacts from facial images, maintaining robust performance even under noisy conditions. In our study, we extract structural descriptions of facial features from LLaVA and encode them using CLIP text encoder to obtain embeddings and then further integrate these embeddings with DRM outputs through a DCA module to guide the image restoration process.

## 3.3 Degradation Removal Module (DRM)

Effective degradation removal is essential for recovering high-fidelity HQ face images from LQ inputs, as corrupted signals can mislead the blind face restoration (BFR) model. To address this, we propose a novel cross-modal mapping module, DRM, which directly transforms degraded face images into noise-suppressed textual representations. This cross-modal strategy is more effective than conventional image-space denoising, as the textual embedding space exhibits a natural decoupling between semantic content and degradation-induced noise.

As shown in Figure 3, the proposed DRM comprises two key components: (1) a Degradation Mapper that projects CLIP image embeddings into implicit textual representations, preserving rich visual semantics that are often lost in explicit textual descriptions; and (2) a Degradation Remover that purifies these representations by filtering out degradation-specific artifacts. The resulting clean textual features maintain high semantic fidelity to the original image content while eliminating noise patterns, thereby providing reliable guidance for subsequent image generation.

**Degradation Mapper.** Following [40], we use a CLIP-based cross-modal projection that maps visual features into a text-aligned embedding space. Specifically, given a degraded input image $X$, we first extract its visual features using a CLIP image encoder $E$, and then project them into the textual embedding space through a learnable Mapper $\mathcal{P}_{\text{mapper}}$:

$$F_{\text{mapper}} = \mathcal{P}_{\text{mapper}}(E(X)), \quad F_{\text{mapper}} \in \mathbb{R}^{N \times D}, \tag{1}$$

where $D$ is the dimensionality of the textual word embeddings, and $N$ is the number of learned tokens (set to 30) to preserve rich visual details while maintaining computational efficiency.

**Degradation Remover.** While $F_{\text{mapper}}$ encodes high-level textual representations, it also carries noise and degradation-specific artifacts that hinder restoration. To address this, we introduce a Degradation Remover $\mathcal{P}_{\text{remover}}$ to purify the token embeddings:

$$F_{\text{remover}} = \mathcal{P}_{\text{remover}}(F_{\text{mapper}}), \quad F_{\text{remover}} \in \mathbb{R}^{N \times D}, \tag{2}$$

where $N$ is consistent with the Mapper design and the cleaned representation $F_{\text{remover}}$ serves as the final conditioning input to the DCA module (Figure 3(c)).

## 3.4 Training and Inference

**Training Stage I: Learning Degradation-Unaware Textual Representations.** The Degradation Mapper projects CLIP image features into a text-aligned embedding space, generating a representation $F_{\text{mapper}}$ that captures both visual content and degradation patterns in a form compatible with text embeddings. As a result, images of different qualities are encoded into a unified textual representation space. During the training of the Degradation Mapper, the condition $F$ is replaced with $F_{\text{mapper}}$ and the training objective is defined as:

$$\mathcal{L}_{\text{stageI}} = \mathbb{E}_{z_0, \epsilon \sim \mathcal{N}(0, I), t} \left[ \| \epsilon - \epsilon_\theta(z_t, t, F) \|_2^2 \right], \tag{3}$$

where $\mathbf{z}_0 = \mathcal{E}(X)$ is the latent representation of input image $X$, encoded by a pretrained VAE encoder $\mathcal{E}(\cdot)$, and $\boldsymbol{\epsilon} \sim \mathcal{N}(0, \mathbf{I})$ represents the added Gaussian noise. At each diffusion timestep $t$, the noisy latent representation $\mathbf{z}_t$ is constructed from $\mathbf{z}_0$ and $\boldsymbol{\epsilon}$ via the forward diffusion process. The noise prediction model $\epsilon_\theta(\cdot)$ is trained to predict the added noise by minimizing the mean squared error between the ground-truth and predicted noise values. The Degradation Remover is trained using the same diffusion loss $\mathcal{L}_{\text{stageI}}$, but with the condition $F$ replaced by $F_{\text{remover}}$, while keeping the Mapper module frozen. Additional training details are provided in the appendix.

**Training Stage II: Simultaneous Prior Injection and Degradation Removal for BFR.** After finishing training Degradation Mapper and Degradation Remover, we freeze their weights, and train the blind face restoration model using simultaneous prior injection and degradation removal across multiple feature scales. Specifically, two complementary sources of information are utilized: (1) a high-level semantic prior $F_{\text{text}}$ obtained from LLaVA, and (2) a degradation-aware, noise-suppressed embedding $F_{\text{remover}}$ produced by the previous stage. They are simultaneously injected into the diffusion model to enable semantically coherent and visually faithful face reconstruction. The training objective follows the standard noise prediction loss used in latent diffusion models:

$$\mathcal{L}_{\text{stageII-BFR}} = \mathbb{E}_{z_0, \epsilon \sim \mathcal{N}(0,I),t} \left[ \| \epsilon - \epsilon_\theta(z_t, t, DCA(F_{\text{text}}, F_{\text{remover}})) \|_2^2 \right], \quad (4)$$

where $DCA(\cdot)$ denotes the Decoupled Cross Attention mechanism.

**Inference Pipeline.** During inference, our framework processes an LQ face image through two parallel paths: the input is first passed through the frozen Degradation Mapper and Remover to obtain a degradation-purified embedding $F_{\text{remover}}$; simultaneously, the same LQ input is used to generate a text embedding $F_{\text{text}}$ via LLaVA [19]. To ensure high-quality text generation, we preprocess the input with GFPGAN-v1.4 [36] prior to feeding it into LLaVA-v1.5-13B [19].

# 4 Experiments

## 4.1 Experimental Settings

**Training Datasets.** We train all models on the FFHQ dataset [11], which contains 70,000 high-quality face images resized to $512 \times 512$. We follow DiffBIR [18] to generate corresponding low-quality (LQ) images. To obtain semantic guidance, we use LLaVA [19] to generate face description prompts for each image.

**Testing Datasets.** We evaluate our method on one synthetic dataset and three real-world datasets. The synthetic dataset, CelebA-Test [10], comprises 3,000 images sampled from CelebA-HQ, with LQ versions generated using the same degradation pipeline as training. The real-world datasets include LFW-Test [9] (1,711 images, mild degradation), WIDER-Test [42] (970 images, heavy degradation), and SCface [8], which features extreme surveillance degradations such as aliasing, jagged artifacts, low-light noise, and defocus blur. SCface contains 130 subjects captured by five surveillance cameras; for evaluation, we use images taken from a distance of 2.6 meters.

**Compared Methods.** We compare our proposed SPIDER with seven state-of-the-art BFR methods across three categories: (1) StyleGAN-prior methods: GFPGAN [36]; (2) Codebook-prior methods: CodeFormer [52] and DAEFR [30]; (3) Diffusion-based methods: DR2 [37], PGDiff [41], DiffBIR [18] and FaithDiff [2]. All experiments are conducted using official code.

**Evaluation Metrics.** We adopt both reference-based and no-reference image quality metrics for comprehensive evaluation. For synthesized datasets with ground truth, we use PSNR [38], SSIM [38], LPIPS [50], FID [1], as well as no-reference perceptual metrics MANIQA [43] and CLIPIQA [33] to better capture perceptual quality. For real-world datasets without ground truth, we evaluate performance using MANIQA, CLIPIQA, and FID.

**Implementation Details** Both the Degradation Mapper and Remover are trained for one epoch on the FFHQ dataset during Stage I, with a batch size of 4, using the Adam optimizer and a learning rate of $1 \times 10^{-6}$ on a single NVIDIA V100 GPU. Our SPIDER model is built upon the pretrained stable-diffusion-2.1 and trained for 15 epochs on two NVIDIA RTX 4090 GPUs, with a batch size of 192, using the Adam optimizer [12] and a learning rate of $5 \times 10^{-5}$.

## 4.2 Comparisons with State-of-the-Art Methods

Table 1: Quantitative comparison on synthetic dataset of CelebA-Test [10]. The best results are marked in red, and the second best in blue.

| Metrics | Input | GFPGAN [36] | CodeFormer [52] | DAEFR [30] | DR2 [37] | PGDiff [41] | DiffBIR [18] | FaithDiff [2] | SPIDER(Ours) |
|---|---|---|---|---|---|---|---|---|---|
| FID ↓ | 152.64 | 21.33 | 22.58 | 15.55 | 27.75 | 19.82 | 28.48 | 20.92 | 22.57 |
| MANIQA ↑ | 0.1683 | 0.4289 | 0.5062 | 0.5426 | 0.5160 | 0.4658 | 0.6534 | 0.5184 | 0.5834 |
| CLIPIQA ↑ | 0.2403 | 0.5391 | 0.6828 | 0.6769 | 0.5972 | 0.5583 | 0.7648 | 0.6570 | 0.7013 |
| LPIPS ↓ | 0.7292 | 0.4554 | 0.3312 | 0.4153 | 0.3354 | 0.3286 | 0.3882 | 0.3145 | 0.3177 |
| PSNR ↑ | 22.56 | 17.86 | 22.67 | 21.89 | 22.26 | 21.52 | 22.90 | 22.76 | 22.94 |
| SSIM ↑ | 0.5006 | 0.5407 | 0.5540 | 0.5966 | 0.5854 | 0.5678 | 0.5410 | 0.5782 | 0.6111 |

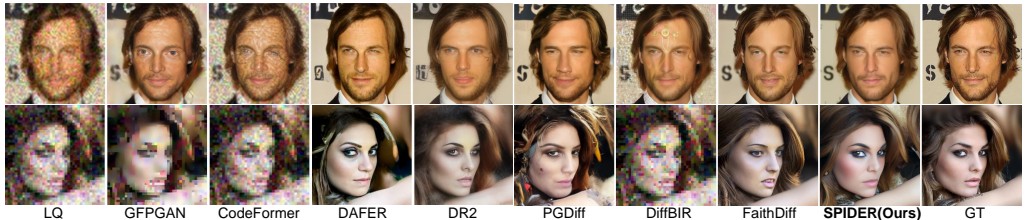

LQ  GFPGAN CodeFormer DAFER  DR2  PGDiff  DiffBIR FaithDiff **SPIDER(Ours)** GT

Figure 4: Qualitative comparison on synthetic dataset of CelebA-Test.

**Evaluation on Synthetic Dataset.** As shown in Table 1, our method achieves state-of-the-art or
near-state-of-the-art performance across multiple evaluation metrics on CelebA-Test, ranking first in
PSNR and SSIM, and second in LPIPS, MANIQA, and CLIPIQA. Figure 4 further demonstrates
that our approach generates perceptually realistic and semantically consistent face restorations, with
well-preserved identity features and effective suppression of visual artifacts. In the first-row example,
clearer eye contours and finer skin details are recovered, while in the second-row case, the original
gaze direction and eye semantics are faithfully maintained—corroborating the superior quantitative
performance.

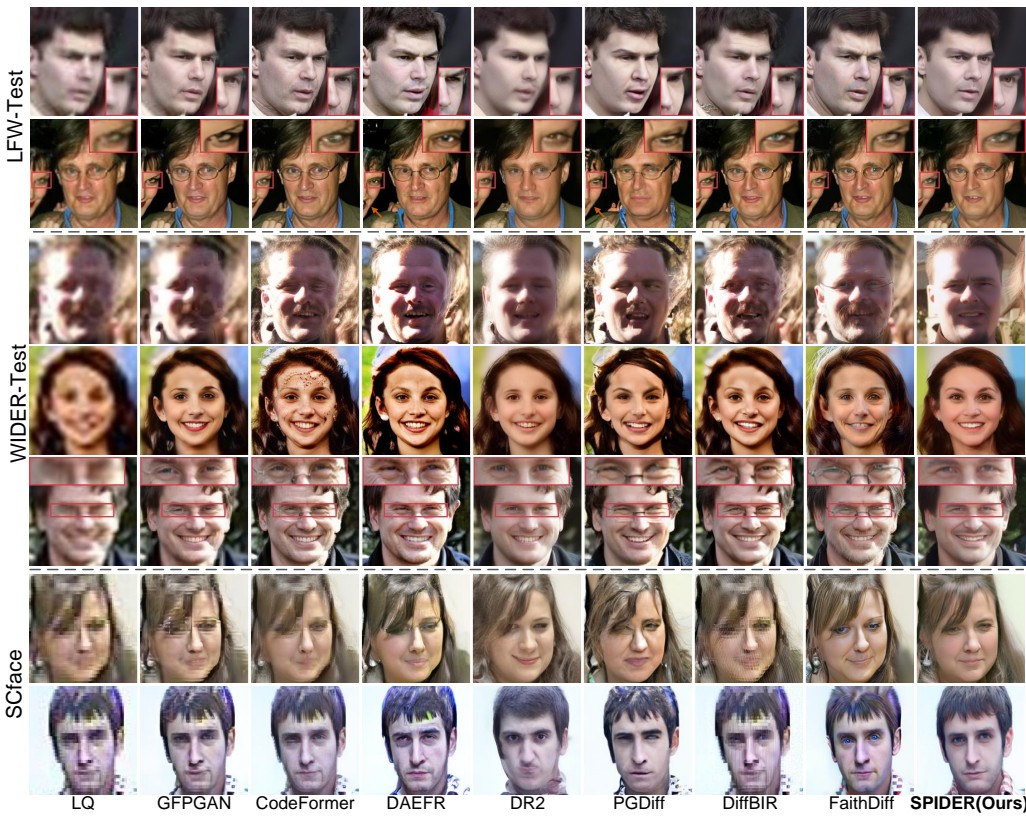

LQ  GFPGAN CodeFormer DAEFR  DR2  PGDiff  DiffBIR FaithDiff **SPIDER(Ours)**

Figure 5: Qualitative comparisons on real-world datasets. Our method is able to restore high quality
faces, showing robustness to the heavy degradation.

**Evaluation on Real-World Dataset.** As shown in Table 2, our method achieved the best FID scores
on the LFW and the WIDER dataset. It also ranked second on both the MANIQA and CLIPIQA
benchmarks. For the SCface dataset, which involves extreme surveillance degradations, our approach
attained the highest MANIQA and CLIPIQA scores. Although the FID score was slightly inferior
to that of PGDiff, our method outperformed it in all other evaluation metrics. Figure 5 presents
qualitative results on real-world datasets. On LFW, our model effectively restores side faces with

clear facial features and minimal background-induced artifacts. Benefiting from the multi-step noise suppression of the DRM module, SPIDER also preserves both primary and background subjects with high fidelity. On WIDER, it maintains key facial features (eyes, nose, mouth) and overall visual consistency, whereas other models suffer from artifacts around facial regions. On SCface, the complex noise patterns inherent in surveillance imagery pose significant challenges for existing models, leading to suboptimal reconstructions. In contrast, our model delivers more faithful and coherent results, demonstrating stronger generalization to real-world degradations.

Table 2: Quantitative comparisons on real-world datasets of LFW [9], WIDER [42], SCface [8]. The best results are marked in red, and the second best in blue.

| Datasets | LFW | | | WIDER | | | SCface | | |
|---|---|---|---|---|---|---|---|---|---|
| Degradation | Mild real-world degradations | | | Heavy real-world degradations | | | Extreme surveillance degradations | | |
| Methods | FID ↓ | MANIQA ↑ | CLIPIQA ↑ | FID ↓ | MANIQA ↑ | CLIPIQA ↑ | FID ↓ | MANIQA ↑ | CLIPIQA ↑ |
| GFPGAN [36] | 53.87 | 0.4558 | 0.6324 | 50.36 | 0.4352 | 0.5756 | 106.97 | 0.4358 | 0.6461 |
| CodeFormer [52] | 52.84 | 0.5266 | 0.6889 | 39.22 | 0.4959 | 0.6984 | 99.07 | 0.4327 | 0.6852 |
| DAEFR [30] | 47.69 | 0.5420 | 0.6965 | 36.72 | 0.5205 | 0.6975 | 103.64 | 0.4600 | 0.7217 |
| DR2 [37] | 50.42 | 0.5248 | 0.6532 | 52.78 | 0.4746 | 0.5948 | 96.49 | 0.4438 | 0.5823 |
| PGDiff [41] | 41.86 | 0.4763 | 0.6070 | 38.06 | 0.4391 | 0.5880 | 85.48 | 0.3610 | 0.5127 |
| DiffBIR [18] | 40.91 | 0.6735 | 0.7948 | 35.82 | 0.6624 | 0.8083 | 149.98 | 0.3965 | 0.4648 |
| FaithDiff [2] | 41.34 | 0.4949 | 0.6787 | 36.07 | 0.5106 | 0.7092 | 88.48 | 0.5127 | 0.6880 |
| **SPIDER(Ours)** | 39.74 | 0.5784 | 0.7320 | 34.58 | 0.5630 | 0.7342 | 87.57 | 0.5514 | 0.7446 |

## 4.3 Ablation studies

**Effectiveness of the Degradation Removal Module.** As shown in Table 3, integrating DRM results in a clear performance improvement across all metrics. A visual comparison on WIDER-Test is presented in Figure 6 (a). In the first and third columns, without DRM, the BFR model misinterprets noise as authentic detail, leading to severe distortions in the hair, facial region, clothing and background. In the second and fourth columns,

Table 3: Ablation results showing the effectiveness of the DRM on the CelebA-Test.

| Metrics | Without DRM | With DRM |
|---|---|---|
| PSNR↑ | 20.76 | 22.94 |
| SSIM↑ | 0.5671 | 0.6111 |
| LPIPS↓ | 0.3910 | 0.3177 |
| FID↓ | 24.94 | 22.57 |

unclear contours and artifacts are generated because the BFR model struggles to effectively distinguish between noise and informative content. These results suggest that the module effectively suppresses noise while preserving fine-grained structural and textural details, thereby enhancing overall image fidelity.

**Insights into SPIDER Design.** Table 4 shows that the order of the semantic prior injection and the DRM is critical. We adopt a design where the semantic prior is injected before DRM, allowing the diffusion model to amplify both signal and noise. This amplification helps DRM better distinguish informative structures from degradation, leading to improved restoration quality. As shown in Figure 6(b), applying DRM first may suppress useful details, resulting in a generated image that lacks detail and exhibits unnatural textures.

Table 4: Ablation results comparing different module orders on the CelebA-Test.

| Metrics | DRM→Semantic prior | Semantic prior→DRM |
|---|---|---|
| PSNR↑ | 21.50 | 22.94 |
| SSIM↑ | 0.5988 | 0.6111 |
| LPIPS↓ | 0.3550 | 0.3177 |
| FID↓ | 27.54 | 22.57 |

**Importance of Face-oriented Prompt Design.** Our method customizes the prompts fed into LLaVA [19] based on CelebA-Test facial attribute definitions [10], enabling richer semantic descriptions of facial structures. According to Table 5, compared to general image descriptions, incorporating detailed facial descriptions can enhance the performance of restoration. Thus, guiding the vision-language model to attend to facial attributes is essential for effectively leveraging its

Table 5: Ablation results comparing different prompt styles on the CelebA-Test.

| Metrics | General Prompt | Facial Attributes Prompt | Face Description Prompt |
|---|---|---|---|
| PSNR↑ | 21.38 | 21.42 | 22.94 |
| SSIM↑ | 0.5899 | 0.5854 | 0.6111 |
| LPIPS↓ | 0.3616 | 0.3744 | 0.3177 |
| FID↓ | 24.57 | 30.50 | 22.57 |

prior knowledge. However, facial attributes prompts often omit spatial information and frequently include redundant features (e.g., "normal nose," "average eyes") that are shared across many images, offering limited benefit for recovering fine textures. More prompt examples are in the appendix.

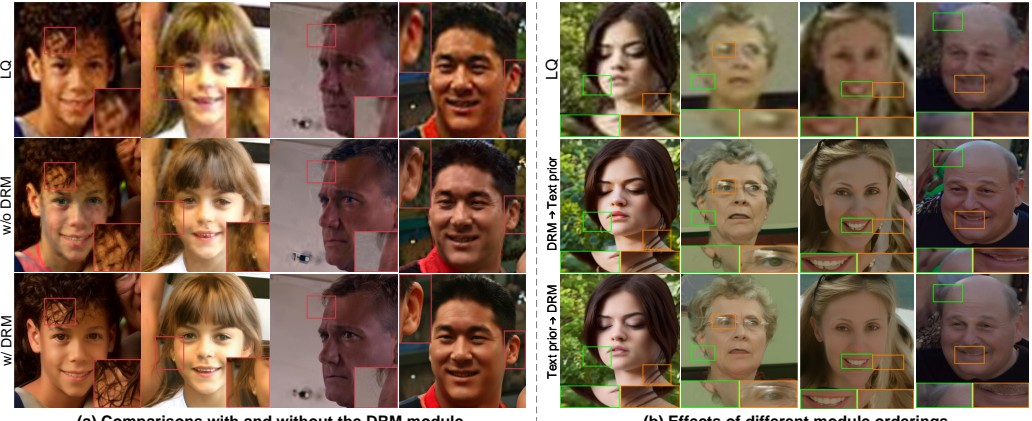

**(a) Comparisons with and without the DRM module**      **(b) Effects of different module orderings**

Figure 6: Ablation studies evaluating (a) the impact of the DRM module and (b) the ordering of modules within the model architecture.

### 4.4 Understanding the Degradation Removal Mechanism in SPIDER

We use t-SNE [32] to visualize feature changes after training the Degradation Mapper and Remover in SPIDER. We randomly select 100 CelebA-Test images (seed=42), synthesize their LQ versions at FID 150 and 250 using the same degradation pipeline as training, and apply the same process to training data (FFHQ) for reference. As shown in Figure 7, we observe that (1) features extracted by the Mapper are widely dispersed, with HQ and LQ clearly separated, indicating strong degradation sensitivity; (2) after applying the Remover, features across all quality levels converge into a degradation-invariant space, where even severely degraded images align closely with HQ ones. These consistent patterns across both training and testing datasets demonstrate that our DRM effectively suppresses degradation-related variations, enabling robust and faithful restoration under extreme degradations.

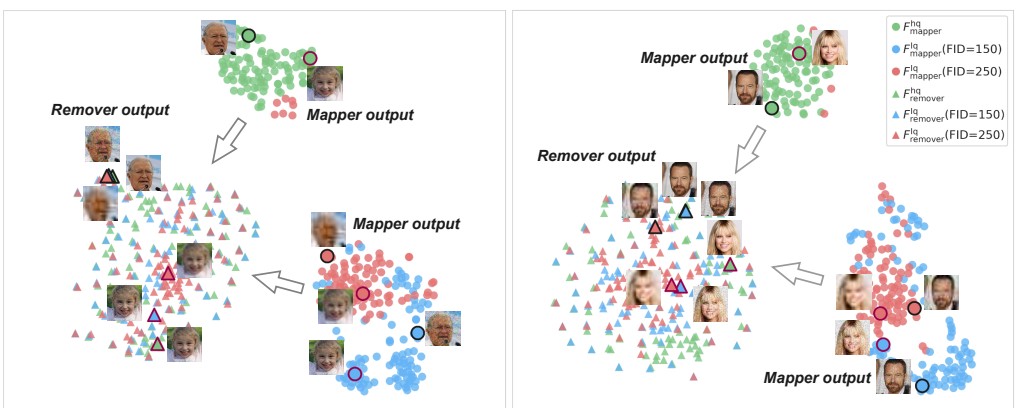

Figure 7: t-SNE visualization of feature distributions on FFHQ (left) and CelebA-HQ (right).

### 5 Conclusion

We propose SPIDER, a novel paradigm for blind face restoration that performs simultaneous multi-level prior injection and degradation removal. SPIDER adopts an interleaved architecture where prior injection precedes degradation removal at each level, ensuring that semantic and diffusion priors amplify both signal and noise in a way that helps the DRM better distinguish meaningful facial features from heavy degradations. Experiments on synthetic and real-world datasets confirm its superiority over state-of-the-art methods. Beyond its strong performance in BFR, SPIDER presents a novel learning paradigm with broad applicability to various image restoration tasks.

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
