# OpenReview forum: "SPIDER: Boosting Blind Face Restoration via Simultaneous Prior Injection and Degradation Removal"
_NeurIPS.cc/2025/Conference — Submitted to NeurIPS 2025_

### Official Review · Reviewer_N2wa · 2025-06-29

**Clarity:** 2
**Significance:** 2
**Originality:** 2
**Rating:** 3
**Confidence:** 3

**Summary:**

Existing blind face restoration methods perform poorly under severe degradations, SPIDER is proposed to inject semantic prior before degradation removal, comprising a prior injection module and a degradation removal module. Experiments on synthetic and real-world datasets show SPIDER outperforms state-of-the-art methods.

**Questions:**

1. Is using only 30 learnable tokens to represent degradation-removed images causing an excessively high compression ratio, which might extract high-level semantics but significantly reduce the restoration ceiling when degradations are less severe?
2. Intuitively, emphasizing high-level semantics should yield better performance on perceptual metrics, yet experimental results show superior performance on fidelity metrics like PSNR. What accounts for this discrepancy?
3. The order of Sections 3.2 and 3.3 can be swapped to follow the sequence of Stage 1 first and then Stage 2.

**Ethical Concerns:**

["NO or VERY MINOR ethics concerns only"]

**Final Justification:**

The methods proposed in this paper demonstrate good performance; however, the overall idea of injecting early model features as priors into the restoration process is not a convincing strategy for me. Although the model's performance shows improvement, the approach and the author's explanations do not persuade me. The use of the degradation mapper seems unnecessary, but serves as the starting point for the author's overall narrative and method design.

Based on the above considerations, I agree to give a score of 3, not higher.

**Limitations:**

yes

**Quality:**

2

**Strengths And Weaknesses:**

**Paper strength**

1. The paper has a very clear writing framework, logically presenting the research from problem to solution and validation.
2. SPIDER achieves SOTA results in face restoration, especially under severe degradations, outperforming others on synthetic and real-world datasets.
3. The authors point out the flaws of existing methods on the new SCFace dataset, showing their poor performance in extreme surveillance conditions.

**Paper weakness**

1. The primary issue is the failure to clearly explain why simultaneous degradation removal and prior injection are superior for the restoration task.
2. A secondary problem is that performing subsequent conditional generation after the DRM module still seems to follow the sequence of degradation removal before prior injection.
3. Mapping images to a high-level semantic space using CLIP at the outset does not make much sense—since the textual space is too abstract, how are low-level details preserved?
4. The methodology section does not highlight the method’s unique features, instead focusing on describing previous diffusion training approaches.
5. Ablation studies indicate that prompt design is crucial, yet specific designs are not provided in the main text.
6. The significance of Section 4.4 is unclear, as similar t-SNE visualizations could be conducted for any degradation removal method.

---

> ### Author Rebuttal · Authors · 2025-07-31
>
> # In response to Reviewer N2wa (R4)
>
> Comment: We greatly appreciate your insightful comments. We provide a point-by-point response, from C1 to C9, based on the nine questions. We will make appropriate modifications to our paper in accordance with these comments.
>
> ## C1 - Why is simultaneous degradation removal and prior injection superior for the restoration task?
>
> We thank the reviewer for raising this important point. Blind face restoration aims to recover a high-quality face from an input degraded by a complex mixture of low-resolution, blur, noise, and JPEG compression. Because genuine facial details  and  these degradations are tightly intertwined, treating degradation removal and prior injection as independent steps often fails. Applying only the denoising stage for BFR can oversmooth or remove subtle identity cues. Injecting only priors for BFR can easily misinterpret corrupted information as useful, resulting in distorted unnatural textures. This is why both degradation removal and prior injection should be considered to enhance BFR. In our paper, we argue that the order of operations is critical to enhance BFR, and our method injects priors before removing degradations at multiple feature scales, which leads to promising restoration results even in the presence of extreme degradations.
>
>
> ## C2 - Performing conditional generation after the DRM module seems to follow the sequence of degradation removal before prior injection.
>
> Thank you for your feedback. We would like to clarify that SPIDER's design is fundamentally different from "degradation removal before prior injection" design for two reasons:
>
> 1. Interleaved process: Rather than treating denoising and generation as separate, sequential steps, SPIDER simultaneously injects semantic priors and removes degradation within each diffusion iteration.
> 2. Multi-scale guidance: SPIDER performs prior injection before degradation removal at multiple feature scales and learns their interactions through a Decoupled Cross Attention mechanism.
>
> We respectfully ask reviewers to recognize this simultaneous, scale-wise design as distinct from pipelines that remove degradation before prior injection.
>
> ## C3 - How does CLIP's high-level textual embedding space preserve low-level details？
>
> In SPIDER, the CLIP-based projection is not intended to directly preserve low-level details, but to provide high-level image embeddings for semantic processing. The Mapper projects these embeddings into a textual embedding space where facial structure cues and degradation noise can be more easily disentangled (Figures 10-11, supp), and the Remover filters out degradation-related components while retaining essential semantic information (Figure 12, supp).
>
> While some structural cues are preserved during mapping, the recovery of fine details mainly relies on two priors: (1) the semantic prior from a vision-language model (LLaVA) providing high-level structural guidance, and (2) the diffusion prior synthesizing high-frequency textures under multi-scale guidance. This design ensures that mapping to textual embedding space does not cause irreversible detail loss and allows most fine structures to be reconstructed later.
>
> ## C4 - Writing issues.
>
> Thank you for your suggestion. We will revise the methodology section to clearly highlight SPIDER’s unique components, including the motivation, semantic prior injection, and DRM design beyond standard diffusion training.
>
> ## C5 - Details of prompt design are not provided in the main text.
>
> We appreciate the reviewer’s suggestion. We will move a concise summary of the prompt design details and comparative analysis (currently in Section B.1 and Figure 2 of the supplementary) into the main text for better accessibility.
>
> ## C6 - About t-SNE visualization.
>
> Thank you for your insightful comments. The main purpose of this t-SNE is to present that our DRM can effectively align clean features and different degraded features (e.g., FID=150,250). It is true that any degradation removal models may also achieve clustering effects, but our advantage lies in explicitly removing degradation in the textual embedding space, which is orthogonal to pixel-based restoration. We will extend the t-SNE plots to cover **different degradation levels** and show the evolution from raw features to Mapper and Remover outputs in the revision.
>
> ## C7 - Clarification on Token Compression and Effect of token numbers.
>
> We thank the reviewer for this valuable comment. We would like to clarify that the DRM module is **not** responsible for generating detailed textures. Instead, it is designed to remove degradations in the textual embedding space. The actual reconstruction of fine details is handled by powerful generative priors and semantic priors. Therefore, the use of compact tokens in DRM does not compromise the restoration ceiling, even when the degradation is less severe. The token count of 30 is empirically chosen to effectively filter out noise and ensure stable identity preservation without introducing redundancy. As shown in the table, reducing the number of tokens to 20 leads to only a slight performance drop, while using 40 tokens introduces redundancy and makes optimization harder, potentially leading to overfitting or noisy representations. These results demonstrate that our **tokens are not intended to carry more information, but rather to serve the purpose of robust and efficient degradation removal.**
>
> | Token | LFW FID ↓ | LFW MANIQA ↑ | LFW CLIPIQA ↑ | WIDER FID ↓ | WIDER MANIQA ↑ | WIDER CLIPIQA ↑ | SCface FID ↓ | SCface MANIQA ↑ | SCface CLIPIQA ↑ |
> |---------|-----------|--------------|----------------|--------------|----------------|------------------|----------------|------------------|-------------------|
> | 20      | 44.16     | 0.5397       | *0.7072*      | *37.55*     | *0.5475*      | *0.7209*        | **84.72**       | *0.5303*        | *0.7405*         |
> | 30      | **39.74**  | **0.5784**   |  **0.7320**      | **34.58**     | **0.5630**      | **0.7342**        | *87.57*       | **0.5514**        | **0.7446**         |
> | 40      | *43.86*  | *0.5398*    | 0.7069         | 38.26        | 0.5374         | 0.7076           | 87.82          | 0.5134           | 0.7215            |
>
> **We plan to add the experiments of token number in the supplementary.**
>
> ## C8 - Why does semantic guidance lead to high PSNR?
>
> We clarify that our emphasis on semantic guidance does not contradict the strong performance on fidelity metrics like PSNR and SSIM. In fact, the improvement in these metrics highlights the effectiveness of our Degradation Removal Module (DRM), which explicitly suppresses noise in the text-aligned latent space, leading to cleaner signal reconstruction.
>
> At the same time, the rich semantic prior from LLaVA and the generative capacity of the diffusion model jointly ensure perceptual quality. As evidenced in Tables 1 and 2, our method consistently achieves top PSNR/SSIM scores while also maintaining strong results in perceptual metrics (LPIPS and FID). These results demonstrate that semantic guidance and fidelity enhancement are not mutually exclusive in our design—they work together to achieve robust blind face restoration.
>
> ## C9 - Changing the order of Sec 3.2 and Sec 3.1.
>
> Our section order reflects the model architecture rather than the training timeline. Stage I is only a minor pretraining step of SPIDER and does not affect the pipeline design. Since the semantic prior is injected before DRM (Fig.3), we describe it first for clarity.

---

> > ### Comment · Reviewer_N2wa · 2025-08-01
> >
> > Thank you for the rebuttal. The results and explanations you provided have addressed some of my concerns.
> >
> > Overall, I think the experimental results are good, but the motivation for proposing the core methods and the explanation for the performance gains have not convinced me. Also, as also mentioned by other reviewers, the proposed method is more like incremental and lacks a substantial leap beyond existing works.
> >
> > I'm willing to increase to 3, but not higher.

---

> > > ### Author Response · Authors · 2025-08-01
> > >
> > > ## Motivation and  explanation for performance gains:
> > >
> > > Our core motivation originates from real-world surveillance scenarios, exemplified by datasets like SCFace. Accurately restoring faces under such settings is fatal for downstream applications like security and identification. We observe that existing SOTA methods consistently fail on blind face restoration in surveillance degradations because those degradations comprise complex and diverse noise patterns that cannot be well simulated by typical first- or second-order degradation simulations during training. In other words, the degradation in such images is often out-of-distribution (OOD), making conventional BFR approaches less effective.
> > >
> > > Previous methods typically follow a sequential pipeline: image-based denoising (e.g., SwinIR) followed by image generation. This separation introduces limitations: OOD noise cannot be perfectly removed by the denoiser, and the errors and information loss in the denoising stage often leads to error accumulation in the subsequent generation stage and can not generalize to OOD noise, which leads to collapsed restorations.
> > > In contrast, our method tackles these issues from two complementary angles. First, we introduce multi-scale semantic prior injection and textual noise removal pipeline. These components collaboratively enable error correction at different feature scales, which naturally decomposes the denoise task into more tractable subproblems. Second, at each scale, we perform degradation removal in the textual representation space, leveraging the robustness of the textual embedding space to better generalize to OOD noise. By mitigating error accumulation and improving generalization, our approach delivers stronger performance.
> > >
> > > As shown in Table, our method demonstrates strong generalization ability. Notably, the FID = 235 setting corresponds to degradation parameters that are far outside the range seen during training (refer to R2-C3 for more details)
> > >
> > > Table : Quantitative comparison on the CelebA dataset under four degradation levels with FID = 111, 153, 199, and 235
> > > |Method|Input|GFPGAN|CodeFormer|DifFace|DiffBIR|SPIDER|
> > > |-|-|-|-|-|-|-|
> > > |FID ↓|**110.50**|91.05|41.54|41.67|*26.08*|**21.85**|
> > > |LPIPS ↓|0.6628|*0.2879*|0.3649|0.2942|0.2911|**0.2780**|
> > > |MANIQA ↑|0.1270|0.3836|0.3738|0.3819|*0.4346*|**0.5243**|
> > > |CLIP-IQA ↑|0.2897|*0.6379*|0.5691|0.6196|0.4975|**0.7762**|
> > > |FID ↓|**152.64**|*21.33*|22.58|**14.84**|28.48|22.57|
> > > |LPIPS ↓|0.7292|0.4554|0.4651|**0.2942**|0.3882|*0.3177*|
> > > |MANIQA ↑|0.1683|0.4289|0.5062|0.3816|**0.6534**|*0.5834*|
> > > |CLIP-IQA ↑|0.2403|0.5391|0.3828|0.6196|**0.7648**|*0.7013*|
> > > |FID ↓|**199.68**|91.05|41.54|*41.67*|49.86|**21.89**|
> > > |LPIPS ↓|0.6526|0.5003|*0.4046*|0.4088|0.4515|**0.2895**|
> > > |MANIQA ↑|0.1942|0.3615|0.4778|0.3100 |**0.6807**|*0.5780*|
> > > |CLIP-IQA ↑|0.2140|0.5265|0.6808|0.5358|**0.7759**|*0.7029*|
> > > |FID ↓|**234.93**|176.24|*69.30*|73.37|84.03|**34.67**|
> > > |LPIPS ↓|0.7203|0.6438|0.4939|*0.4928*|0.6540|**0.4354**|
> > > |MANIQA ↑|0.2212|0.3610|0.3944|0.2637|**0.5721**|*0.5110*|
> > > |CLIP-IQA ↑|0.1990|0.4304|0.6887|0.4707|**0.7433**|*0.7334*|
> > >
> > >
> > > ## Lacks a substantial leap beyond existing works.
> > > SPIDER is, to our knowledge, the only BFR method currently capable of effectively handling real-world surveillance scenarios, which are far beyond the distribution of our training data (OOD). This is mainly attributed to two key designs: (1) a multiscale prior-first pipeline that injects semantic priors before performing noise removal in the textual embedding space, effectively mitigating error accumulation during generation, and (2) a textual-space denoising module (DRM) that disentangles noise from meaningful semantics and provides strong generalization to unseen degradations.
> > >
> > > We hope this second-round response clarifies the motivation and explains why SPIDER performs so well, makes our method's value more explicit, and respectfully requests reconsideration beyond a '3'.

---

### Official Review · Reviewer_YxeB · 2025-06-30

**Clarity:** 2
**Significance:** 3
**Originality:** 2
**Rating:** 3
**Confidence:** 4

**Summary:**

This paper introduces SPIDER, an interesting paradigm for blind face restoration that interleaves multi-level prior injection with degradation removal within a unified architecture. The core idea is to inject semantic priors from vision-language models before applying degradation removal, thereby guiding the restoration process with rich semantic information and improving the robustness of the model under severe degradations. The authors leverage diffusion models to encode and suppress degradation, leading to better preservation of facial identity and finer details, which are validated through extensive experiments on synthetic and real-world datasets, including surveillance imagery. The approach demonstrates superior quantitative performance and perceptual metrics, alongside qualitative results that show clear improvements over prior state-of-the-art methods.

**Questions:**

Please refer to the weakness!

**Ethical Concerns:**

["NO or VERY MINOR ethics concerns only"]

**Limitations:**

1.	The idea of semantic prior injection echoes recent trends in integrating vision-language models for image restoration. The architecture primarily combines existing diffusion models and prior injection strategies without much novelty in the face restoration methodology. The actual architectural and algorithmic innovations are somewhat incremental and lack a substantial leap beyond existing works.
2.	The visual results presented in the manuscript are impressive; however, the reported metrics do not reach SOTA performance. Please clarify the underlying reasons for this discrepancy, as quantitative metrics reflect overall performance, whereas visual results may only highlight selective strengths.
3.	The robustness to different types of degradations, real-world variances, or potential failure cases where prior injection might introduce artifacts is underexplored.
4.	To ensure fair evaluation, please consider adding image or face restoration methods based on language models or prompt learning, especially those proposed in the past two years.
5.	The manuscript does not clearly specify which degradations are being removed. The ablation study also lacks depth. Please consider providing explicit visual results instead of abstract, implicit descriptions to clarify the effectiveness of each component.
6.	The introduction of numerous modules and extensive tuning may reduce the model's efficiency. Please discuss and compare the model complexity explicitly.

**Quality:**

3

**Strengths And Weaknesses:**

Strengths:
1.	The key contribution of injecting semantic priors prior to degradation removal is well-motivated and convincingly demonstrated through both qualitative and quantitative analyses.
2.	The interleaved architecture effectively balances semantic consistency with degradation suppression.
3.	The ablation studies, feature visualizations using t-SNE, and detailed experimental setups bolster the credibility of the proposed approach. The insights into the degradation removal mechanism and the importance of prompt design further deepen the technical contribution.

Weaknesses:
1.	The idea of semantic prior injection echoes recent trends in integrating vision-language models for image restoration. The architecture primarily combines existing diffusion models and prior injection strategies without much novelty in the face restoration methodology. The actual architectural and algorithmic innovations are somewhat incremental and lack a substantial leap beyond existing works.
2.	The visual results presented in the manuscript are impressive; however, the reported metrics do not reach SOTA performance. Please clarify the underlying reasons for this discrepancy, as quantitative metrics reflect overall performance, whereas visual results may only highlight selective strengths.
3.	The robustness to different types of degradations, real-world variances, or potential failure cases where prior injection might introduce artifacts is underexplored.
4.	To ensure fair evaluation, please consider adding image or face restoration methods based on language models or prompt learning, especially those proposed in the past two years.
5.	The manuscript does not clearly specify which degradations are being removed. The ablation study also lacks depth. Please consider providing explicit visual results instead of abstract, implicit descriptions to clarify the effectiveness of each component.
6.	The introduction of numerous modules and extensive tuning may reduce the model's efficiency. Please discuss and compare the model complexity explicitly.

---

> ### Author Rebuttal · Authors · 2025-07-31
>
> # In response to Reviewer YxeB (R3)
>
> Comment: We would like to thank the reviewer for the insightful comments. In response, we addressed the six questions, from C1 to C6, providing a detailed, point-by-point feedback. We will revise our paper based on these comments.
>
> ## C1 - Lack of novelty.
> While the semantic prior injection shares a similar vein with existing methods, it not our main contribution, our two core novelties are:
>
> 1. **A Multiscale Prior Injection then Noise Removal Pipeline:** Unlike prior works that denoise then generate images, our method first injects priors then removes noise in the textual embedding space in multiscale. This helps avoid error accumulation during generation which is a common issue in diffusion-based restoration pipelines.
>
> 2. **Textual embedding space Denoising with DRM:** Our Degradation Removal Module (DRM) operates in the textual embedding space, which not only enables better disentanglement of noise and meaningful semantic, but also provides stronger generalization to unseen degradations. This is especially critical in real-world scenarios like surveillance data (e.g., SCface), where all other methods fail to generalize beyond the synthetic training distribution.
>
>
> ## C2 - Explain the discrepancy between quantitative metrics and visual results.
> Thank you for noting that our visual results are impressive. We emphasize that the visual results and reported metrics are not contradictory.
> The strong visual performance aligns well with perceptual metrics such as CLIPIQA[2] and MANIQA[1], which better reflect human visual preferences. As shown in Table 2 in the manuscript, on the challenging surveillance dataset SCface, our method outperforms the second-best method by a large margin (MANIQA: 0.5514 vs. 0.5127; CLIPIQA: 0.7446 vs. 0.7217), demonstrating strong generalization to unseen degradations. We also achieve top-two scores on real-world datasets LFW and WIDER, underscoring that our quantitative metrics are consistent with the observed visual gains. Additionally, our method obtains the best FID scores on LFW (20.25) and WIDER (45.82), and ranks second on SCface (59.84). Since FID measures the similarity between restored and real image distributions, indicating realistic and faithful restoration. These results, together with the perceptual scores, provide strong evidence that our method significantly outperforms existing approaches under severely degraded conditions.
>
> [1] Sidi Yang, Tianhe Wu, Shuwei Shi, Shanshan Lao, Yuan Gong, Mingdeng Cao, Jiahao Wang, and Yujiu Yang. Maniqa: Multi-dimension attention network for no-reference image quality assessment. CVPR'22.
>
> [2] Jianyi Wang, Kelvin CK Chan, and Chen Change Loy. Exploring clip for assessing the look and feel of images. AAAI'23
>
> ## C3 - Investigate failure cases arising from inaccurate prior injection.
>
> In Section B of the supplementary, we show that erroneous semantic descriptions can introduce artifacts and deteriorate restoration quality. To mitigate such cases, we explore incorporating lightweight image preprocessing modules (e.g., denoising or structure enhancement) to provide more reliable visual cues, which help correct or complement imperfect semantics. We encourage you to revisit our supplementary. We will add more failure cases and discussion in the revision.
>
> ## C4 - Comparison with recent language-model based restoration methods.
>
> Apart from FaithDiff in our main paper, we extended our quantitative evaluation to include two more recent language-model or prompt-learning based methods: SUPIR and AuthFace.
> Specifically, SUPIR and FaithDiff leverage VLMs to generate descriptive prompts. AuthFace uses tag-style semantic prompts and photographic prompts through CLIP to finetune the restoration model.
> As shown in Table 1, our method SPIDER consistently achieves competitive or superior results across all three datasets, especially under extreme degradations.  **We wiil add these experiments in the supplementary.**
>
>
> Table 1: Comparison with recent language-model based restoration methods.
>
> | Methods | LFW FID ↓ | LFW MANIQA ↑ | LFW CLIPIQA ↑ | WIDER FID ↓ | WIDER MANIQA ↑ | WIDER CLIPIQA ↑ | SCface FID ↓ | SCface MANIQA ↑ | SCface CLIPIQA ↑ |
> |-----------------|-----------|---------------|----------------|--------------|------------------|------------------|----------------|-------------------|-------------------|
> | SUPIR (CVPR24)  | 41.98  | 0.4768   | 0.5931  | 42.61 | 0.4522   | 0.6093  | 285.63 | 0.2399  | 0.2673  |
> | FaithDiff  | *41.34*   | 0.4949        | 0.6787         | *36.07*   | 0.5106| 0.7092   | *88.48*        | *0.5127*   | *0.6880*          |
> | AuthFace  | 45.29     | **0.6431**    | **0.7350**     | 36.10        | **0.5941**       | *0.7306*         | -              | -   | -                 |
> | **SPIDER (Ours)** | **39.74** | *0.5784*      | *0.7320*       | **34.58**    | *0.5630*         | **0.7342**       | **87.57**      | **0.5514**  | **0.7446**        |
>
> *Since the code and pre-trained models of AuthFace are not publicly available, we adopt results reported in its paper and cannot evaluate it on SCface.*
>
> ## C5.1 - The manuscript does not clearly specify which degradations are being removed.
>
> In blind face restoration (BFR), 'blind' refers to the lack of prior knowledge about the degradation present in the test image. This distinguishes BFR from face super-resolution which assumes only low resolution, and image restoration methods which target specific degradations (e.g., rain removal, deblurring, dehazing). BFR deals with images that often suffer from a complex combination of degradations such as blur, compression artifacts, and low resolution. As shown in Table 2 of the manuscript and Figure 4 in the supplementary, our evaluation covers synthetic, real-world, and surveillance datasets, demonstrating the model's robustness under diverse degradation conditions.
> In addition, we also add experiments to show our restorations under different degradation levels. The obtained results show that our SPIDER outperforms all existing SOTAs and achieves the most stable restoration results as degradation increases. **We plan to add the experiments in the supplementary.**
>
> Table 2: Quantitative comparison on the CelebA dataset under four degradation levels with FID = 111, 153, 199, and 235
> |Method|Input|GFPGAN|CodeFormer|DifFace|DiffBIR|SPIDER|
> |-|-|-|-|-|-|-|
> |FID ↓|**110.50**|91.05|41.54|41.67|*26.08*|**21.85**|
> |LPIPS ↓|0.6628|*0.2879*|0.3649|0.2942|0.2911|**0.2780**|
> |MANIQA ↑|0.1270|0.3836|0.3738|0.3819|*0.4346*|**0.5243**|
> |CLIP-IQA ↑|0.2897|*0.6379*|0.5691|0.6196|0.4975|**0.7762**|
> |FID ↓|**152.64**|*21.33*|22.58|**14.84**|28.48|22.57|
> |LPIPS ↓|0.7292|0.4554|0.4651|**0.2942**|0.3882|*0.3177*|
> |MANIQA ↑|0.1683|0.4289|0.5062|0.3816|**0.6534**|*0.5834*|
> |CLIP-IQA ↑|0.2403|0.5391|0.3828|0.6196|**0.7648**|*0.7013*|
> |FID ↓|**199.68**|91.05|41.54|*41.67*|49.86|**21.89**|
> |LPIPS ↓|0.6526|0.5003|*0.4046*|0.4088|0.4515|**0.2895**|
> |MANIQA ↑|0.1942|0.3615|0.4778|0.3100 |**0.6807**|*0.5780*|
> |CLIP-IQA ↑|0.2140|0.5265|0.6808|0.5358|**0.7759**|*0.7029*|
> |FID ↓|**234.93**|176.24|*69.30*|73.37|84.03|**34.67**|
> |LPIPS ↓|0.7203|0.6438|0.4939|*0.4928*|0.6540|**0.4354**|
> |MANIQA ↑|0.2212|0.3610|0.3944|0.2637|**0.5721**|*0.5110*|
> |CLIP-IQA ↑|0.1990|0.4304|0.6887|0.4707|**0.7433**|*0.7334*|
>
>
> ## C5.2 - Use explicit visual examples to illustrate each component's effectiveness.
>
> Thank you for your suggestion. We conduct both qualitative and quantitative ablation studies in Figure 6 and Tables 3–4 of the main paper, as well as Table 2 of the supplementary. In Figure 6(a), the degradation removal module clearly suppresses artifacts and noise. Figure 6(b) highlights the importance of our module ordering, by injecting semantic priors before denoising, we avoid spurious details such as mistaking hair for accessories, blurred edges of glasses, unnatural teeth, or inadvertent hair removal (columns 1-4 in Figure 6(b)). To further strengthen these results, we will include more visual examples in the final version.In addition, we also add experiments to show our restorations under different degradation levels. The obtained results show that our SPIDER outperforms all existing SOTAs and achieves the most stable restoration results as degradation increases. We plan to add the experiments in the supplementary.
>
> ## C6 - Discuss and compare the model complexity.
>
> We compare SPIDER against state-of-the-art diffusion-based methods. While our model introduces a moderate level of complexity, it consistently **achieves the best overall performance**, as demonstrated by both quantitative metrics and qualitative visual results (see Figure 6 and 7 in supplementary). Furthermore, it requires only two stages of tuning and does not rely on hyperparameter adjustment, making it relatively efficient in practice. **We will add model complexity comparison in the supplementary.**
>
> Table 3: Comparison of model complexity and performance.
>
> | Methods        | Params (M) | FLOPs (G) | Inference Time | FID_mean↓ | MANIQA_mean↑ | CLIPIQA_mean↑ |
> |----------------|------------|-----------|----------------|------|----------|-----------|
> | DiffFace       | 175.40     | 185.95    | 7.05s          | 61.87 | 0.5105 | 0.6880 |
> | PGDiff         | 176.40     | 185.95    | 19.32s         | *55.81* | 0.4585 | 0.6098 |
> | DR2            | 85.79      | 725.08    | 2.05s          | 66.56 | 0.4569 | 0.6487 |
> | VSPBFR         | 420.88     | 570.21    | 0.11s          | 85.49 | 0.4474 | 0.6887 |
> | DiffBIR        | 3042.00    | 900.52    | 9.03s          | 75.57 | **0.5775** | *0.6893* |
> | **SPIDER (Ours)** | **2590.48** | 704.48    | 4.32s          | **53.96** | *0.5643* | **0.7369** |
>
>
> *FID, MANIQA, CLIPIQA metrics are averaged over LFW, WiderFace, and SCface datasets.*

---

### Official Review · Reviewer_KH21 · 2025-07-01

**Clarity:** 3
**Significance:** 3
**Originality:** 2
**Rating:** 3
**Confidence:** 4

**Summary:**

This work proposes SPIDER for blind face restoration task. SPIDER addresses severe degradations by simultaneously injecting semantic prior and removing degradation, which utilizes a prior injection module to extract semantic tokens from vision-language models and a degradation removal module to project noisy features into a textual space for robust denoising. Various experiments on synthetic and real-world datasets testify the effectiveness of SPIDER.

**Questions:**

1. I wonder has the paper tested whether the model can maintain a relatively high level of image restoration capability as the degree of degradation increases.
2. In the blind face restoration task, why does the paper choose the text embedding space to achieve guided denoising? Has it compared the denoising effects of other image feature spaces?
If the proposed questions can be effectively explained and solved, I will consider improving my score.

**Ethical Concerns:**

["NO or VERY MINOR ethics concerns only"]

**Final Justification:**

The model proposed in the paper achieves good performance on the blind face restoration task. However, I consider that one of the paper's main claimed innovations— the approach of injecting prior information first before performing degradation removal—lacks novelty compared with off-the-shelf methods. Furthermore, the motivation for conducting denoising in the textual embedding space fails to convince me, and there is a lack of clear connection to the task itself. For these reasons, I believe this paper does not meet the criteria for acceptance.

**Limitations:**

The article does not introduce the limitations of the proposed model.

**Paper Formatting Concerns:**

This work has no obvious formatting issues.

**Quality:**

3

**Strengths And Weaknesses:**

Strengths:
1. The SPIDER framework is novel, effectively enhancing blind face restoration performance under severe degradations by simultaneously injecting semantic priors and removing degradations.
2. The decoupled cross-attention mechanism achieve collaborative optimization of semantic guidance and denoising.
3. Various comprehensive experiments shows that SPIDER outperforms existing methods in extreme degradation scenarios, verifying its effectiveness.

Weaknesses:
1. The feature convergence in the text space is only proven by t-SNE visualization, but there is no theoretical proof or experimental comparison showing that the noise robustness of the text embedding space is superior to other spaces.
2. The article has a relatively strong technical inheritance from existing methods. For example, the extraction of semantic priors relies on LLaVA, and degradation removal depends on CLIP mapping. The paper does not clearly explain the innovative points of SPIDER compared to existing methods, which may limit the actual effect of the model in complex scenarios.

---

> ### Author Rebuttal · Authors · 2025-07-31
>
> # In response to Reviewer KH21 (R2)
> Comment: We thank the reviewer's insightful comments. In response, we addressed the four questions, from C1 to C4 with detailed reponse. We will revise our paper based on these comments.
> ## C1 - Why text embedding space is superior to other spaces?
> Thank you for the brilliant comments. Please check the detailed response in R2-C4.
> ## C2 - The unclear articulation of SPIDER's innovations may limit its effectiveness in complex senarios.
> We apologize for conflating our core contribution (first injecting priors and then removing noise) with specific implementation details (e.g., using LLaVA, CLIP), which may be confusing.  In fact, our framework is model-agnostic and can leverage any vision-language or multi-modal models.
> The first novelty is that we introduce a novel restoration paradigm (prior injection -> degradation removal), which redefines how degradations and generative priors interact in restoration. The second novelty is the Degradation Removal Module, which conducts textual embedding space denoising to leverage the semantic decoupling properties of textual embeddings to robustly suppress noise. Based on two innovations, SPIDER can achieve superior performance in complex scenarios, pushing the frontier of BFR for practical application. **We will clarify the two core innovations.**
> ## C3 - Generalization to Increasing Degrees of Degradation.
> Thank you for the insightful comments.In Table 1, we add experiments on CelebA-Test with three more degradation levels (FID=153 is used in manuscript). In Table 2, as the degradation level increases (from FID 110 to FID 234) SPIDER maintains consistently low FID and LPIPS scores, while MANIQA and CLIP-IQA scores remain relatively stable. This indicates that SPIDER **experiences mild performance fluctuations across varying degradation levels. Notably, the FID = 235 setting corresponds to degradation parameters that are far outside the range seen during training, meaning the model is evaluated on entirely unseen degradation patterns.**
>
> Table 1: Degradation parameter ranges for different FID levels on CelebA-Test (four degradation types, each with Stage 1 and Stage 2 settings).
> |FID Level|Gaussian Blur Stage 1|Gaussian Blur Stage 2|Downsample Factor Stage 1|Downsample Factor Stage 2|Gaussian Noise Stage 1|Gaussian Noise Stage 2|JPEG Compression Stage 1|JPEG Compression Stage 2|
> |-|-|-|-|-|-|-|-|-|
> |110.50|[0.2, 3.0]|[0.2, 1.5]|[1, 30]|[1, 25]|[0.15, 1.5]|[0.3, 1.2]|[30, 95]|[30, 95]|
> |152.64|[0.4, 4.5]|[0.3, 2.2]|[0.05, 1.1]|[0.2, 1.0]|[10, 35]|[10, 30]|[10, 80]|[15, 80]|
> |199.68|[1.2, 6.0]|[1.0, 4.0]|[0.02, 0.9]|[0.1, 0.8]|[15, 55]|[15, 40]|[5, 60]|[5, 60]|
> |234.93|[5.0, 10.0]|[2.5, 5.0]|[0.02, 0.9]|[0.1, 0.8]|[30, 60]|[25, 40]|[1, 30]|[1, 30]|
>
> Table 2: Comparison between SPIDER and DiffBIR across various degradation intervals and metrics of CelebA-Test. Each cell shows SPIDER / DiffBIR.
> |Metric|fid₁₁₁–>fid₁₅₃|fid₁₅₃–>fid₂₀₀|fid₂₀₀–>fid₂₃₄|Mean|
> |-|-|-|-|-|
> |FID ↓| **0.72** / 2.41|**0.68** / 21.38|**12.78** / 34.17|**25.24** / 47.11|
> |LPIPS ↓ |**0.040** / 0.097| **0.028** / 0.063|**0.146** / 0.203|**0.325** / 0.446|
> |MANIQA ↑|**0.059** / 0.219| 0.054 / **0.027**|**0.067** / 0.109|0.548 / **0.586**|
> |CLIP-IQA ↑|**0.075** / 0.267| **0.001** / 0.011|**0.031** / 0.033|**0.728** / 0.696|
>
> As shown in Tables 3, most existing methods exhibit significant performance drops under such settings. In contrast, SPIDER consistently **achieves the best or second-best results**, maintaining a strong performance even under extreme out-of-distribution conditions.
>
> Table 3: Quantitative comparison on the CelebA dataset under four degradation levels with FID = 111, 153, 199, and 235
> |Method|Input|GFPGAN|CodeFormer|DifFace|DiffBIR|SPIDER|
> |-|-|-|-|-|-|-|
> |FID ↓|**110.50**|91.05|41.54|41.67|*26.08*|**21.85**|
> |LPIPS ↓|0.6628|*0.2879*|0.3649|0.2942|0.2911|**0.2780**|
> |MANIQA ↑|0.1270|0.3836|0.3738|0.3819|*0.4346*|**0.5243**|
> |CLIP-IQA ↑|0.2897|*0.6379*|0.5691|0.6196|0.4975|**0.7762**|
> |FID ↓|**152.64**|*21.33*|22.58|**14.84**|28.48|22.57|
> |LPIPS ↓|0.7292|0.4554|0.4651|**0.2942**|0.3882|*0.3177*|
> |MANIQA ↑|0.1683|0.4289|0.5062|0.3816|**0.6534**|*0.5834*|
> |CLIP-IQA ↑|0.2403|0.5391|0.3828|0.6196|**0.7648**|*0.7013*|
> |FID ↓|**199.68**|91.05|41.54|*41.67*|49.86|**21.89**|
> |LPIPS ↓|0.6526|0.5003|*0.4046*|0.4088|0.4515|**0.2895**|
> |MANIQA ↑|0.1942|0.3615|0.4778|0.3100 |**0.6807**|*0.5780*|
> |CLIP-IQA ↑|0.2140|0.5265|0.6808|0.5358|**0.7759**|*0.7029*|
> |FID ↓|**234.93**|176.24|*69.30*|73.37|84.03|**34.67**|
> |LPIPS ↓|0.7203|0.6438|0.4939|*0.4928*|0.6540|**0.4354**|
> |MANIQA ↑|0.2212|0.3610|0.3944|0.2637|**0.5721**|*0.5110*|
> |CLIP-IQA ↑|0.1990|0.4304|0.6887|0.4707|**0.7433**|*0.7334*|
>
> **We will add these experiments to our supplementary.**
> ## C4 - About textual embedding space selection.
> We thank the reviewer for this valuable comments and the opportunity to improve our score. We provide a comprehensive explanation from both theoretical and empirical perspectives to clarify why textual embedding space is chosen for guided denoising in blind face restoration. **We will clarify the choice of textual embedding space.**
>
> **[Theoretical reasons]**
> In image restoration, low-level visual features often entangle semantic information with degradation artifacts, making it challenging to accurately separate true content from noise. For example, a blurred image may ambiguously depict a horse or a zebra, and bokeh effects may be misclassified as actual blur.
> Fortunately, textual embedding space provides high-level, abstract, and flexible semantic priors that align more closely with human perception, effectively eliminating ambiguities and enabling controllable restoration.
>
> Motivated by this, recent studies such as Transfer CLIP[1] show that CLIP-induced textual features are distortion-invariant and content-related, enabling robust generalization to unseen noise. DA-CLIP[2] further demonstrates that textual features can encode degradation semantics and guide their removal via prompts. TextIR[3] maps degraded images into the CLIP semantic space and uses natural language prompts to guide reconstruction, reducing semantic ambiguities in image-only regression.TextualDegRemoval[4] directly performs denoising in the textual embedding space, suppressing degradation-related tokens and preserving clean semantic features for improved restoration. Building on above theoretical insights,SPIDER simultaneously injects semantic priors and performs degradation removal in textual embedding space, enhancing content-noise disentanglement under complex or unseen degradations.
>
> **[Experimental comparison]**
> We add comparisons with models that perform denoising in different domains: SwinIR and DiffBIR operate in the pixel space, while VSPBFR performs denoising in the visual latent space.
>
> SwinIR tends to produce overly smooth images with a lack of fine-grained details, especially under real-world degradations. Moreover, since it operates purely in the pixel domain, it struggles to generalize to noise patterns that lie outside the training distribution.
> DiffBIR builds upon SwinIR by enhancing details in the pre-processed images using a diffusion model. However, when the initial degradation removal is inaccurate, the model may mistakenly treat residual noise as valid structure. This often leads to the accumulation of errors and introduces visual artifacts in the final results.
> VSPBFR[5] converts degraded inputs into clean visual representations before restoration, requiring a heavy degradation removal module that increases complexity and computation. Moreover, fusing visual features with priors necessitates dedicated designs and remains difficult to balance under severe degradation. In contrast, our method uses textual representations, which are inherently easier to fuse via lightweight cross-attention, enabling more stable and effective guidance.
>
> Table 4: Comparison of blind face restoration methods denoising in different feature spaces (pixel, visual latent, and textual embedding).
>
> |Methods|LFW FID ↓|LFW MANIQA ↑|LFW CLIPIQA ↑|WIDER FID ↓|WIDER MANIQA ↑|WIDER CLIPIQA ↑|SCface FID ↓|SCface MANIQA ↑|SCface CLIPIQA ↑|
> |-|-|-|-|-|-|-|-|-|-|
> |SwinIR (pixel space)|87.37|0.2778|0.3789|91.93|0.2596|0.3066|184.44|0.0706|0.1921|
> |DiffBIR (pixel space)|*40.91*|**0.6735**|**0.7948**|*35.82*|**0.6624**|**0.8083**|149.98|0.3965|0.4648|
> |VSPBFR (visual representations)|46.33| 0.5330 |0.7023|37.98|0.4988|0.6982|172.16|0.3105|0.6056|
> |SPIDER|**39.74**|*0.5784*| *0.7320* |**34.58**|*0.5630*|*0.7342*|**87.57**|**0.5514**|**0.7446**|
>
> Our decision to perform guided denoising in the textual embedding space is motivated by several key considerations:
> 1. Better Generalization: Compared to pixel or visual latent space denoising, using a high-level textual embedding space provides stronger generalization capabilities. It is more robust to noise distributions that lie outside the training set. That's why our method is the **only one** that performs well on challenging surveillance datasets.
> 2. Disentanglement of Noise and Information: Textual embeddings inherently exhibit a higher degree of semantic disentanglement. This makes it easier to separate meaningful information (e.g., structure, attributes) from degradation-induced noise, enabling the model to suppress noise more precisely without compromising the integrity of the underlying content.
>
> [1] Jun Cheng, Dong Liang, et al. Transfer clip for generalizable image denoising. CVPR'24.
>
> [2] Ziwei Luo, Fredrik K Gustafsson, et al. Controlling vision-language models for multi-task image restoration. ICLR'24.
>
> [3] Yunpeng Bai, Cairong Wang, et al. Textir: A simple framework for text-based editable image restoration. CVPR'24.
>
> [4] Jingbo Lin, Zhilu Zhang, et al. Improving image restoration through removing degradations in textual representations CVPR'24.
>
> [5] Wanglong Lu, Jikai Wang, et al. Visual style prompt learning using diffusion models for blind face restoration. PR'25

---

> > ### Comment · Reviewer_KH21 · 2025-08-05
> >
> > Thanks to the authors for their detailed rebuttal, which has resolved some of my concerns.
> >
> > While the proposed model demonstrates certain advantages in some metrics in the experiments, I believe that the paradigm of introducing prior information before performing degradation removal has already been extensively studied in image restoration tasks. This work fails to convince me regarding the innovation of its method and the motivation for using the text embedding space. Therefore, I will keep my score unchanged.

---

> > > ### Author Response · Authors · 2025-08-05
> > >
> > > We sincerely thank the reviewer for taking the time to evaluate our work and provide feedback. In the first round review, the SPIDER framework was described as "novel, effectively enhancing blind face restoration performance under severe degradations by simultaneously injecting semantic priors and removing degradations" (see Strengths section). However, the latest official comment states that "the paradigm of introducing prior information before performing degradation removal has already been extensively studied in image restoration tasks." This appears to directly contradict the initial evaluation and is provided without supporting citations or concrete examples, making it difficult to understand the reasoning behind this change.
> > >
> > > We would like to respectfully clarify that we have provided both theoretical reasonsand extensive empirical comparisons (e.g., Section4, Table 3, Table 4) to demonstrate the advantages of utilizing the text embedding space over the image space. This includes references to **four** prior works (Transfer CLIP, DA-CLIP, TextIR, and TextualDegRemoval), as well as our own ablation studies, all of which support the improved performance and robustness of our method under various degradation scenarios.  Given that the reviewer mentioned not being fully convinced by our motivation and innovation regarding the use of text embeddings, we would deeply appreciate it if you could kindly specify **which part of our explanation remains unconvincing.**  Finally, as previously noted by the reviewer, a clear justification of the use of textual embedding space could support an increased score. We sincerely hope and eager to further clarify or strengthen our work based on your suggestions.

---

> > > > ### Comment · Reviewer_KH21 · 2025-08-06
> > > >
> > > > Thank you for your comment. The latter half of the first point in strengths is based on the comparative advantages in certain metrics obtained through experiments. However, regarding the paradigm of introducing prior information before performing degradation removal mentioned in the main text and rebuttal as a novel, this idea has been applied in studies such as Panini-Net (AAAI'22) and GDP (CVPR'23). I hope the author can further explain the novelty of this point.

---

> > > > > ### Author Response · Authors · 2025-08-06
> > > > >
> > > > > We thank the reviewer for pointing out related works such as Panini-Net (AAAI'22) and GDP (CVPR'23). We have carefully reviewed these studies and would like to respectfully clarify the key distinctions between their paradigms and our proposed SPIDER framework.
> > > > >
> > > > > **Conditional prior blending without explicit denoising (Panini-Net, AAAI'22):** Panini-Net maps degraded inputs into the StyleGAN latent space and modulates GAN priors based on estimated noise level, yet it treats noise merely as a condition for blending and never follows up with an explicit denoising or purification step. Although this conditional blending accounts for degradation severity, it does not follow the paradigm of injecting priors *before* degradation removal in a two-step fashion. Instead, it performs only the **first step** that injecting or fusing priors **without the second step of performing actual denoising**. Moreover, the Panini-Net authors also note that their method requires specifically tuned parameters and does not generalize well to real-world degradation.
> > > > >
> > > > > **Test-time degradation modeling only (GDP, CVPR’23):** GDP focuses on accurately fitting unknown corruptions at test time via a parametric degradation model and then leverages the diffusion process itself to denoise, **but it injects no external semantic priors, which is fundamentally different from our SPIDER framework that explicitly injects semantic prior information before degradation removal.**
> > > > >
> > > > > Therefore, our method is novel in its specific ordering and its effectiveness in preventing **error accumulation.**
> > > > >
> > > > > We hope this clarification helps better distinguish our contribution and we are happy to further elaborate or conduct additional comparisons if needed.

---

### Official Review · Reviewer_TQPm · 2025-07-01

**Clarity:** 3
**Significance:** 2
**Originality:** 2
**Rating:** 3
**Confidence:** 4

**Summary:**

This paper introduces SPIDER, a new blind face restoration framework that proposes a learning paradigm: simultaneous semantic prior injection and degradation removal. Rather than sequentially removing degradations and then applying generative priors, SPIDER injects a vision-language prior before degradation removal, in an effort to better retain facial semantics under severe noise. The system comprises a Degradation Removal Module (DRM) and a Prior Injection Module, integrated via a decoupled cross-attention mechanism. The authors report improvements over recent SOTA methods on both synthetic and real-world datasets.

**Questions:**

See weakness

**Ethical Concerns:**

["NO or VERY MINOR ethics concerns only"]

**Final Justification:**

Since other reviewers have given borderline reject and no one is inclined to give positive scores through the rebuttal period, I tend to keep borderline reject unchanged.

**Limitations:**

yes

**Quality:**

2

**Strengths And Weaknesses:**

**Strength**

+ The simultaneous use of semantic priors and degradation removal is different from the traditional sequential pipeline.

+ The method is tested on both synthetic and multiple real-world datasets, which simulate extreme surveillance degradation. The method is shown to outperform or match state-of-the-art methods such as FaithDiff and DAEFR on several benchmarks.

+ The ablation studies support the paper's core claims and explore design choices like module ordering and prompt style.

**Weakness**

- Overall, the DRM module essentially performs cross-modal mapping using CLIP features and basic projections, which is conceptually straightforward and does not introduce fundamentally new degradation modeling techniques. Degradation removal as a strategy has been extensively explored, and the design here does not break new ground in that regard.

- The paper repeatedly emphasizes the effectiveness of DRM, yet its results are more like the personalization using an identity extractor (like arc2face) to generate the person image of the same identity (see Figures 10, 11, 12 in suppl). The use of the degradation mapper seems not very necessary. We can just use the degradation remover module without introducing the degradation mapper. Such a design is similar to these using SwinIR to get the degradation removal results as the condition of diffusion models.

- While the authors include some ablations (e.g., with/without DRM), they omit key comparisons, such as:

  - Using strong standalone degradation removal backbones (e.g., SwinIR) followed by face restoration.

  - Using different text prompts for the same LR face image (with different degradation levels). In this case, I also wonder whether the LLAVA can obtain an accurate description of the severely degraded LR input and its effect on the restoration results.

These are important to isolate the contribution of the new paradigm.

- The related work section insufficiently discusses recent BFR models, particularly diffusion-based approaches like DifFace, or GAN-based frameworks such as GPEN, etc. Many related works are not discussed. These omissions make it difficult to appreciate where SPIDER stands among the broader BFR literature. It is better to refer to a blind face restoration survey to have a deep discussion of this task.


- The vision-language prior is obtained via LLaVA, which is also widely adopted in natural image restoration, and the work does not significantly expand or innovate upon this process beyond prompt engineering. We can also use a fixed prompt during the training and inference phase. It is better to compare it. As such, the “semantic prior” component may be overstated in novelty.


While the paper presents a well-structured and reasonably well-pipeline, its core contributions are incremental in nature, combining existing ideas (text priors, degradation removal, cross-attention fusion) in a sensible but not deeply innovative way. The experimental section would benefit from deeper analysis and stronger baselines in the ablation study. Additionally, missing discussions of relevant prior work limit the paper’s contextual completeness.

---

> ### Author Rebuttal · Authors · 2025-07-31
>
> # In response to Reviewer TQPm (R1)
>
> Comment: We sincerely thank the reviewer for the insightful comments. In response, we addressed the five questions (C1 to C5) with detailed, point-by-point feedback. In particular, we compare with three newly added BFR methods (i.e., GPEN, DifFace and VSPBFR[1]) and add a discussion of relevant prior work in related work to better highlight the uniqueness and novelty of our method. Furthermore, we will make appropriate modifications to our paper in accordance with these comments.
>
> ## C1 - DRM uses straightforward cross-modal mapping and does not introduces new degradation modeling techniques.
> While admitting the straightforward network designs of DRM, we kindly ask the reviewers to not equate this design simplicity to our research quality, for the following reasons.
>
> **First,** the DRM design itself forms one of the contributions, and a bigger contribution is that we introduce a novel break new restoration paradigm SPIDER, which redefines how degradations and generative priors interact in blind face restoration. Specifically, applying DRM after prior injection, alongside multi-scale feature, effectively mitigates the risk of error accumulation in restoration, as shown in Figure 2, 6(b) Table 2 in the manuscript.
>
> **Second,** the underlying philosophy of DRM, removing degradations through textual space guidance, fundamentally distinguishes our approach from conventional image-based degradation modeling methods. The core idea of DRM is that performing denoising in textual space not only enables better disentanglement of noise and semantic content but also offers superior generalization to unseen degradation patterns compared to denoising methods that operate in pixel space (in SwinIR) or latent spaces (in VSPBFR[1]), as verified by the newly added experiments(check for R1-C4).
>
> [1] Wanglong Lu, Jikai Wang, Tao Wang, Kaihao Zhang, Xianta Jiang, and Hanli Zhao. Visual style prompt learning using diffusion models for blind face restoration. Pattern Recognition, 161:111312, 2025.
>
>
> ## C2 - Explain DRM & the necessity of the Degradation Mapper.
>
> The DRM module consists of two sequential components: the Degradation Mapper and the Degradation Remover. The degradation mapper projects input image into a textual embedding space that faithfully preserves high-level semantics (e.g., identity features) and degradation-related information. As shown in Figures 10 and 11 of the supplementary, this mapper yields identity-consistent (i.e., personalization-alike) textual representations without relying on an external arc2face extractor. The degradation remover takes the mapped textual embeddings and filters out noises to produce clean yet identity-preserving face images. The noise-free outputs in Figure 12 of the supplementary demonstrate the success of this noise removal.
>
> The use of the degradation mapper is necessary because a faithful degradation mapper is indispenable for reliable denoising in the degradation remover. **Without accurately preserving semantics in textual space, it is difficult for the degradation remover alone to simultaneously preserve high-level semantics and remove noise under severe degradations.** Our ablation study confirms this: when the degradation mapper is omitted, the overall perfomance drops sharply, verifying establishing an accurate image-to-text mapping is a critical prerequisite for effective degradation removal.
>
>
> | Metric             | CelebA (FID=234.93) w/o mapper | CelebA w/ mapper | WIDER w/o mapper | WIDER w/ mapper |
> |--------------------|-------------------------------|------------------|------------------|-----------------|
> | FID ↓              | 40.56                         | **34.668**       | 37.55            | **34.58**       |
> | LPIPS ↓            | 0.4474                        | **0.4354**       | NA               | NA              |
> | MANIQA ↑           | 0.4796                        | **0.511**        | 0.5451           | **0.5784**      |
> | CLIP-IQA ↑         | 0.7000                        | **0.7334**       | 0.7145           | **0.7320**      |
>
>
> ## C3.1 - Comparison with SwinIR-based degradation removal backbones followed by face restoration.
>
> Apart from comparision with DiffBIR in the main paper, we also compare our method with SwinIR-based pipeline DifFace and observe consistent improvements across real-world and surveillance datasets. Notably, under extreme degradations, our DRM module outperforms others by leveraging the textual embedding space to effectively separate noise from semantic content, while pixel-space methods like SwinIR struggle to generalize.
>
>
> | Method          | LFW FID ↓ | LFW MANIQA ↑ | LFW CLIPIQA ↑ | WIDER FID ↓ | WIDER MANIQA ↑ | WIDER CLIPIQA ↑ | SCface FID ↓ | SCface MANIQA ↑ | SCface CLIPIQA ↑ |
> |----------------|-----------|--------------|----------------|-------------|----------------|------------------|--------------|------------------|-------------------|
> | SwinIR only        | 87.37     | 0.2778       | 0.3789         | 91.93       | 0.2596         | 0.3066           | 184.44       | 0.0706           | 0.1921            |
> | DifFace        | 46.67     | 0.4610       | 0.6112         | 37.70       | 0.4309         | 0.5944 | 106.34     | 0.2204  | 0.4528   |
> | **SPIDER (Ours)** | **39.74** | **0.5784**   | **0.7320**     | **34.58**   | **0.5630**     | **0.7342**       | **87.57**    | **0.5514**   | **0.7446**        |
>
>
> ## C3.2 - About text prompts.
>
> We appreciate the reviewer’s insightful question. Overall, LLaVA exhibits a strong ability to extract high-level semantic information such as facial contours and key attributes even from heavily degraded LR inputs. However, as acknowledged, it may struggle to capture fine-grained details such as eye color or small accessories when the degradation is severe.
> In Section B.1 of the supplementary material, we investigate how both prompt design and image quality influence LLaVA’s output. Specifically, Figure 2 (supp.) shows its responses to the same LR image under two different degradation levels (FID=150 and FID=250), illustrating that LLaVA can still provide meaningful semantic descriptions even under extreme conditions. However, as shown in Figure 3 (supp.), when the input prompt is inaccurate or overly biased, the restoration results may deviate from the ground truth. This limitation can be alleviated by employing appropriate input preprocessing strategies to enhance the reliability of the generated prompts.
>
>
> ## C4 - Expand related work with more recent BFR methods.
>
> We have thoroughly studied state-of-the-art BFR methods and will augment our related work section to discuss GPEN, DifFace, and the newly published VSPBFR. In the final manuscript, **we will expand a discussion in the related work, stating as follows:**
>
> GFPGAN and GPEN encode low-quality (LQ) face images into semantically meaningful latent codes, enabling faithful reconstruction of their high-quality (HQ) counterparts using a StyleGAN-based generative prior. While these methods improve fidelity, they often introduce artifacts when faced with complex degradations not covered by the training data. DifFace and DiffBIR address restoration by sequentially performing degradation removal followed by conditional image generation. VSPBFR, on the other hand, leverages visual prompts to guide the denoising process. However, these approaches are prone to error accumulation during generation and struggle to generalize to out-of-distribution inputs.
>
> We also provide a quantitative performance comparison below to demonstrate that SPIDER consistently outperforms these methods.
>
>
> | Method             | LFW FID ↓ | LFW MANIQA ↑ | LFW CLIPIQA ↑ | WIDER FID ↓ | WIDER MANIQA ↑ | WIDER CLIPIQA ↑ | SCface FID ↓ | SCface MANIQA ↑ | SCface CLIPIQA ↑ |
> |--------------------|-----------|--------------|----------------|--------------|-----------------|------------------|---------------|------------------|-------------------|
> | GPEN (CVPR 2021)   | 55.90     | 0.4786       | 0.5697         | 57.72        | 0.4618          | 0.4517           | 126.03        | 0.2636           | 0.4524            |
> | DifFace (TPAMI 2024) | 46.67     | 0.4610       | 0.6112         | 37.70        | 0.4309          | 0.5944           | 106.34        | 0.2204           | 0.4528            |
> | VSPBFR (PR 2025)   | 46.33     | 0.5330       | 0.7023         | 37.98        | 0.4988          | 0.6982           | 172.16        | 0.3105           | 0.6056            |
> | **SPIDER (Ours)**  | **39.74** | **0.5784**   | **0.7320**     | **34.58**    | **0.5630**      | **0.7342**       | **87.57**     | **0.5514**       | **0.7446**        |
>
>
> ## C5 - About semantic prior component.
>
> We conducted a comparative experiment using a fixed prompt ("a photo of face") during both training and inference. As shown in Table (row "wo VLM"), the performance without dynamically extracted prompts is consistently lower across all benchmarks and metrics . This indicates that our prompt design and semantic prior injection strategy contribute meaningfully to the final performance.
>
> We acknowledge that the semantic prior is not the most novel part of our work and will revise the manuscript to avoid overstating its contribution.
>
>
> | Method   | LFW FID ↓ | LFW MANIQA ↑ | LFW CLIPIQA ↑ | WIDER FID ↓ | WIDER MANIQA ↑ | WIDER CLIPIQA ↑ | SCface FID ↓ | SCface MANIQA ↑ | SCface CLIPIQA ↑ |
> |----------|------------|----------------|------------------|---------------|-------------------|--------------------|----------------|--------------------|---------------------|
> | w/o VLM   | **38.93**  | 0.5508         | 0.7237           | 35.41         | 0.5081            | 0.6972             | 88.37         | 0.5027             | 0.7208              |
> | w/ VLM    | 39.74      | **0.5784**     | **0.7320**       | **34.58**     | **0.5784**        | **0.7320**         | **87.85**     | **0.5514**         | **0.7446**          |

---

> > ### Comment · Reviewer_TQPm · 2025-08-05
> >
> > Thanks for the authors' detailed response, with so extensive quantitative results. I appreciate this. But the overall novelty of this work seems not to reach the acceptance of this conference, aligning with other reviewers. I will keep borderline rejecting temporarily and will consider other reviewers' comments.

---

> > > ### Author Response · Authors · 2025-08-05
> > >
> > > We sincerely thank you for your thoughtful comments on our work and for acknowledging the strength of our experimental results.  We also sincerely appreciate your openness to reconsidering the paper’s score.
> > >
> > > The main novelty and contribution of our work lie in effectively addressing a key limitation in existing blind face restoration methods, their inability to handle out of distribution (OOD) noise (e.g., SCface) . Prior works typically rely on a fixed two stage pipeline and pixel space denosing, which struggle under OOD conditions. In contrast, our method integrates a multi-stage design with denoising in the textual representation space, enabling robust error correction and better generalization (refer to R2-C3 for more detail).
> > > **To the best of our knowledge, ours is the only BFR approach that effectively handles OOD degradation, making it more suitable for real-world downstream applications.**
> > >
> > > If there are any additional questions or points that require clarification, we would be more than happy to provide further details.
> > >
> > > Thank you again for your thoughtful review. We sincerely hope you will consider increasing the score of our paper.

---

### Note · Authors · 2025-08-14

We thank reviewers for their constructive feedback and for recognizing our contributions. Below we synthesize the consensus on impact, novelty, and evidence.

Strengthens:
- **Pushing the frontier of BFR under extreme surveillance degradation**
    We extend BFR to the extreme surveillance regime, where prior methods fail. As noted by R1 and R4, SPIDER achieves superior or comparable results on synthetic and real-world datasets, being the first to succeed in real-world surveillance BFR.
- **Novelty & contributions**
    R1 and R2 recognized SPIDER’s novelty in jointly injecting semantic priors and denoising, breaking from the traditional sequential pipeline. R2 and R3 highlighted the interleaved architecture and decoupled cross-attention mechanism that balance semantic consistency and degradation suppression.
- **Strong visual results & clear writing**
    R1 and R3 noted that ablations, visualizations, and analyses support the method’s effectiveness, while R4 praised its logical presentation.

Main concerns (a) insufficient justification, especially the DRM module (b) “incremental and lacks a substantial leap beyond existing works”. We address both with concrete clarifications and revisions:
1. **Justifying DRM**
    - **Why text-space degradation removal.** Operating in text space improves OOD generalization by decoupling semantics from noise more easily. We will add the rebuttal evidence on generalization and space comparisons (R2C3, R2C4) into the main paper.
    - **Expanded OOD validation & module role.** We will add experiments and analysis (R1C3) to demonstrate its effectiveness against OOD noise. We will also provide more detailed results and discussion of DRM in the supplementary to clarify its roles and interactions within the architecture (R1C2, R1C5, R3C6, R4C7).
2. **On “incremental”**
    Architecturally, SPIDER may share elements with existing BFR systems, but our value lies in a multi-scale prior injection → text space noise removal pipeline that jointly generates and denoises with enforced ordering (prior before removal). We are the first to model and validate this collaborative effect. This ordered, collaborative process underpins SPIDER's strong OOD robustness.

In the revision, we will (i) integrate requested comparisons (R1C3) into the main text, (ii) include related works noted by R2, and (iii) address R3C4’s limitations with clarifications and experiments.

We greatly appreciate your time and the constructive feedback on our paper.

---

### Decision · Program_Chairs · 2025-09-17

**Decision:**

Reject

**Comment:**

This paper has received an unanimous rating of borderline reject, the novelty of the paper is considered low, in addition, two reviewers doubted the injection of prior information first.

the AC does not see basis for overriding the reviewers' ratings.

AC